# Therapeutic Potential of Quercetin, Silibinin, and Crocetin in a High-Fat Diet-Induced Mouse Model of MASLD: The Role of CD36 and PLIN3

**DOI:** 10.3390/life15101523

**Published:** 2025-09-26

**Authors:** Maria Sotiropoulou, Ioannis Katsaros, Michail Vailas, Fotini Papachristou, Paraskevi Papakyriakopoulou, Nikolaos Kostomitsopoulos, Alexandra Giatromanolaki, Georgia Valsami, Alexandra Tsaroucha, Dimitrios Schizas

**Affiliations:** 1First Department of Surgery, National and Kapodistrian University of Athens, Laikon General Hospital, 11527 Athens, Greece; marosotiropoulou@gmail.com (M.S.); mike_vailas@yahoo.com (M.V.); schizasad@gmail.com (D.S.); 2Laboratory of Experimental Surgery, Faculty of Medicine, Democritus University of Thrace, 68100 Alexandroupolis, Greece; fpapachr@med.duth.gr (F.P.); atsarouc@med.duth.gr (A.T.); 3Laboratory of Biopharmaceutics-Pharmacokinetics, Department of Pharmacy, School of Health Sciences, National and Kapodistrian University of Athens, 15772 Athens, Greece; ppapakyr@pharm.uoa.gr (P.P.); valsami@pharm.uoa.gr (G.V.); 4Laboratory Animal Facility, Biomedical Research Foundation of the Academy of Athens, 11527 Athens, Greece; nkostom@bioacademy.gr; 5Department of Pathology, University Hospital of Alexandroupolis, Democritus University of Thrace, 68100 Alexandroupolis, Greece; agiatrom@med.duth.gr

**Keywords:** MASLD, NAFLD, hepatic steatosis, lipid metabolism, phytochemicals, quercetin, silibinin, crocetin, CD36, PLIN3

## Abstract

Background: Metabolic dysfunction-associated steatotic liver disease (MASLD) is a prevalent and progressive liver disorder linked to metabolic syndrome affecting over 30% of global population, currently lacking effective pharmacological treatment. Natural compounds like quercetin, silibinin, and crocetin have shown hepatoprotective potential. This study investigates their therapeutic effect in a high-fat diet (HFD)-induced mouse model of MASLD. Methods: Ninety-five C57BL/6J (wild type) mice were fed an HFD for 12 weeks to induce hepatic steatosis and were then randomized into eight groups for a 4-week therapeutic intervention. Liver histopathology was assessed using the NAFLD Activity Score (NAS), and immunohistochemistry was conducted to quantify CD36 and PLIN3 expressions. Results: Both quercetin groups significantly reduced the prevalence of steatohepatitis (*p*-value < 0.05) and showed an increased PLIN3 expression. Silibinin also improved steatohepatitis, with the high-dose group reaching statistical significance (*p*-value 0.020), and demonstrated upregulation of PLIN3 along with significant CD36 downregulation. Crocetin groups markedly improved disease severity and showed the highest PLIN3 expression, though without significant changes in CD36. Conclusions: Quercetin, silibinin, and crocetin mitigate MASLD progression by reducing steatohepatitis. These effects are associated with distinct modulations of CD36 and PLIN3 protein expression, suggesting that these pathways are promising therapeutic targets in MASLD management. Natural compounds offer a multi-targeted hepatoprotective approach warranting further clinical investigation.

## 1. Introduction

Metabolic dysfunction-associated steatotic liver disease (MASLD), previously referred to as non-alcoholic fatty liver disease (NAFLD), is a hepatic disorder marked by the accumulation of fat in the liver, occurring in conjunction with at least one cardiometabolic risk factor, such as obesity, hypertension, impaired glucose tolerance, or dyslipidaemia [1,2,3,4]. MASLD has emerged as the most prevalent cause of chronic liver disease globally, affecting more than one-third of the adult population and ranking as the second leading indication for liver transplantation in the United States [5,6]. MASLD has a global prevalence of up to 30% and is projected to impact over 100 million individuals in the Western world by 2030 [7]. Sedentary behaviors, insufficient physical activity, and unhealthy dietary patterns are key contributors to metabolic dysfunction and the rising prevalence of this condition [8].

MASLD encompasses a spectrum of histological stages, beginning with simple hepatic steatosis, which can progress to metabolic dysfunction-associated steatohepatitis (MASH), formerly known as non-alcoholic steatohepatitis (NASH). MASH is characterized by lobular inflammation, hepatocellular ballooning, fibrosis, and, in severe cases, cirrhosis, which significantly elevates the risk of hepatocellular carcinoma [5]. While life expectancy is normal with simple steatosis, MASH significantly increases mortality from liver-related and cardiovascular complications [9,10,11]. The disease’s pathophysiology is now understood by a “multiple parallel hits” hypothesis, where factors like insulin resistance, oxidative stress, and gut microbiota alterations simultaneously cause liver injury, replacing the older “two-hit” model [3,12]. Early MASLD is reversible with lifestyle changes, but advanced stages can cause irreversible damage [13]. Although no definitive pharmacological treatment has been established, numerous natural compounds have demonstrated potential efficacy in managing MASLD.

Natural polyphenols have been employed in managing metabolic disorders due to their well-documented hepatoprotective effects [14]. These natural polyphenolic flavonoids demonstrate anti-inflammatory, antioxidant, anti-apoptotic, and immunoprotective properties [15]. While numerous studies have confirmed their hepatoprotective effects, the precise pathophysiological mechanisms underlying their impact on MASLD remain incompletely elucidated and warrant further research [16,17]. These compounds may help restore redox homeostasis by directly neutralizing excess reactive oxygen species (ROS) or by preventing the dysregulation and inactivation of endogenous antioxidant enzymes [18,19]. Furthermore, phytochemicals were shown to improve cardiovascular function by reducing chronic low-grade inflammation [20]. They decrease insulin resistance and decrease hepatic lipid accumulation resulting in intrahepatic steatosis improvement [21]. These beneficial effects are largely attributed to their ability to promote lipid metabolism in the liver and inhibit fibrotic lipolysis, which helps alleviate oxidative stress and prevent damage to hepatocytes [22]. Clinical trials have also shown their potential in humans by lowering inflammation, NAFLD activity score and liver biochemical parameters [23,24,25]. These compounds also lower liver fat content and also steatosis [24].

Lipophagy is a fundamental process essential for preventing intracellular degeneration and is pivotal in sustaining liver homeostasis [26]. It governs hepatocellular lipid metabolism and the regulation of intracellular lipid reserves, with its activation exerting a hepatoprotective effect [27]. Numerous studies have demonstrated that hepatic autophagy is impaired in established MASLD, although the underlying pathophysiological mechanisms remain incompletely understood [28,29]. Differentiated cluster 36 (CD36), a scavenger receptor class B protein, functions as a membrane receptor capable of binding various lipid and non-lipid ligands. It is widely acknowledged that CD36 plays a significant role in hepatic steatosis by facilitating fatty acid uptake, triglyceride storage, and secretion [30]. Although no definitive studies have established how CD36 inhibitors may halt the progression of MASLD, and no CD36 inhibitors have been clinically approved for MASLD treatment, CD36 remains a promising target for further research. Recently, the potential of natural products targeting CD36 as therapeutic agents for MASLD has garnered considerable interest [30]. Conversely, Perilipin 3 (PLIN3) is a lipid droplet-associated protein that plays a significant role in hepatic lipid metabolism [31]. PLIN3 is involved in the formation and stabilization of lipid droplets (LDs) within hepatocytes, contributing to the regulation of intracellular lipid storage. In the context of MASLD, studies have shown that PLIN3 expression is upregulated in the liver, correlating with increased lipid accumulation and steatosis [32].

This study aimed to assess the effect of natural compounds on MASLD and to explore the role of key autophagy related proteins (CD36, PLIN-3) in the disease’s progression and potential therapeutic modulation. More specifically, we assessed the effect of (1) quercetin and silibinin when administered in the form of their lyophilized products with hydroxypropyl-β-cyclodextrin, as QUE-HP-βCD and SLB-HP-β-CD, respectively, and (2) crocetin, the main active metabolite of crocin, which is the primary constituent of the well-known natural product saffron, when administered in the form of the lyophilized aqueous saffron extract (SFE).

## 2. Materials and Methods

### 2.1. Experimental Model

This study was carried in the animal facility of the Centre of Clinical, Experimental Surgery and Translational Research of the Biomedical Research Foundation of the Academy of Athens. An ethical approval was obtained by the Veterinary Authorities of Region of Athens, Greece (Ethical approval num. no. 792333/4.12.2019) and the Bioethics Committee of National and Kapodistrian University of Athens (no. 239/28.1.2020).

Ninety-five male C57BL/6J (wild type) mice were fed high fat diet (HFD) (21% fat, 0.15% cholesterol and 19.5% caesin, Harland, Teklad, TD88137) for 12 weeks in order to establish fatty liver disease, as established by Christodoulou et al. [33]. The mice were kept in a specific pathogen-free (SPF) facility using individually ventilated cages (Tecniplast, Varese, Italy). The holding room was maintained on a 12 h light/dark cycle with an ambient temperature of 22 ± 2 °C and a relative humidity of 45 ± 10%. Afterwards, the mice were randomly divided into eight groups (Figure 1) to receive by oral gavage on the following interventions; (a). baseline group (n = 11) that did not receive any intervention and were euthanatized under deep sleep anesthesia; (b). control group (n = 12) that had HFD for extra 4 weeks; (c). quercetin low-dose group (QUE LD, n = 12) that had HFD and 200 μL of the QUE-HP-βCD lyophilized product reconstituted in water for injection (WFI) (dose equivalent to quercetin 10 mg/kg or 0.25 mg, based on the content of QUE in the lyophilized product) for extra 4 weeks; (d). quercetin high-dose (QUE HD, n = 12) that had HFD and 200 μL of the QUE-HP-βCD lyophilized product reconstituted in WFI (dose equivalent to quercetin 50 mg/kg or 1.25 mg) for extra 4 weeks; (e). silibinin low-dose group (SLB LD, n = 12) that had HFD and 200 μL of the SLB-HP-βCD lyophilized product reconstituted in WFI at a dose equivalent to silibinin 25 mg/kg or 0.625 mg, based on the content of SLB in the lyophilized product) for extra 4 weeks; (f). silibinin high-dose (SLB HD, n = 12) that had HFD and 200 μL of the SLB-HP-βCD lyophilized product reconstituted in WFI (dose equivalent to silibinin 50 mg/kg or 1.25 mg based on the content of SLB in the lyophilized product) for extra 4 weeks; (g). crocetin low-dose group (CRO LD, n = 12) that had HFD and 200 μL of the lyophilized SFE reconstituted in WFI (dose of 30 mg/kg or 0.75 mg of SFE and equivalent to crocetin 2.84 mg/kg or 0.071 mg, based on the content of SFE in crocin and using the MW of crocin and crocetin to convert crocin to crocetin that is the hydrolysis product and active metabolite in vivo) for extra 4 weeks; (h). crocetin high-dose (CRO HD, n = 12) that had HFD and the lyophilized SFE reconstituted in WFI at a dose of 90 mg/kg (equivalent to crocetin 8.51 mg/kg or 213 mg, based on the content of SFE in crocin and using the MW of crocin and crocetin to convert crocin to crocetin that is the main hydrolysis product and active metabolite in vivo) for extra 4 weeks. After the experimental period, all mice were euthanatized under deep anesthesia. Histopathological evaluation of livers followed using NAFLD activity score (NAS). Furthermore, CD36 and PLIN-3 expressions were also evaluated using immunohistochemistry.

Dose selection for the three natural products was based and clearly documented in our previous published work, as well as in relevant literature studies [13,33,34,35,36,37,38].

### 2.2. Chemicals and Reagents

Quercetin (MW 302.24 g/mol), silibinin (MW 482.44 g/mol) and hydroxypropyl-β-cyclodextrin (HP-β-CD; MW 1460 g/mol) were obtained from Sigma-Aldrich (St. Louis, MO, USA) and Ashland (Covington, KY, USA), respectively. Stigmas of Crocus sativus L. were kindly donated by the Association of Crocus Producers in Kozani, Greece. All preparations utilized HPLC-grade water purchased from Fischer Scientific, Portsmouth, NH, USA.

### 2.3. Preparation of QUE-HP-β-CD and SLB-HP-β-CD Lyophilized Product

Lyophilized powders of Que-HP-β-CD and SLB-HP-β-CD were prepared with molar ratios of 1:2 by freeze-drying aqueous solutions of Que-HP-β-CD and SLB-HP-β-CD, respectively, following the method described by Manta et al. [39]. Briefly, the appropriate amount of HP-β-CD was placed in a 600 mL beaker and suspended in 500 mL of water. Then, 500 mg of either QUE or SLB was added under continuous stirring and protected from light in the case of QUE due to molecule’s photosensitivity. Small amounts of 6% (*v*/*v*) ammonium hydroxide were gradually added until complete dissolution of QUE or SLB, while maintaining the pH at approximately 9.0–9.5 or 10–10.5, respectively. The resulting solutions were transferred into round trays, frozen at −73 °C, and freeze-dried using a Vacuum Freeze Dryer [BK-FD10T, Biobase Biodustry Co. Ltd., Shandong, China)]. The QUE and SLB contents in the lyophilized powders were quantified by high-performance liquid chromatography (HPLC), following the methods described by Papakyriakopoulou et al. and Christodoulou et al. [38,40]. The QUE content in the lyophilized product was determined to be 7.3 ± 0.15% (*w*/*w*), while the SLB content was 11.11 ± 0.68% (*w*/*w*).

### 2.4. Preparation of Saffron Aqueous Extract

Preparation of the saffron extract (SFE) was carried out using the method described by Christodoulou et al. [33]. Briefly, 1 g of saffron stigmas was mixed with 37.5 mL of HPLC-grade water in a glass bottle and stored in a refrigerator for 3 days. The resulting extract was then filtered under low pressure and frozen, followed by lyophilization, using the above-described conditions. The extract contained 27.8 (±0.1)% *w*/*w* of crocin, the primary active constituent of SFE (equivalent to 9.34% *w*/*w* of crocetin, the main hydrolysis product and active metabolite of crocin in vivo) and 0.34 (±0.02)% *w*/*w* of safranal, as determined by HPLC-PDA. The purity of the saffron stigmas used was ensured, as they had been previously authenticated and chemically characterized by chromatographic and spectroscopic analyses, confirming the identity of major constituents and the absence of significant interfering compounds [33].

### 2.5. NAFLD Activity Score

The “NAFLD Activity Score—NAS” was introduced in 2005 by NASH Clinical Research Network to evaluate NAFLD grade [36]. It has three components that are combined for the total score: steatosis (score 0−3), lobular inflammation (score 0−3) and hepatocytes ballooning (score 0−2). A steatosis greater than 5% is a prerequisite for the establishment of NAFLD diagnosis [36]. A NAS equal to or greater than 5 is indicative of steatohepatitis, whereas a score less than 5 is considered as inconclusive [36].

### 2.6. Immunohistochemistry (Methods)

Immunohistochemistry analysis was performed blindly by an expert pathologist utilizing Nikon Eclipse 50i microscope and Basler acA1920-40uc camera. The pylon Viewer, a tool within the Basler pylon v. 25.09 Software Suite for Windows, Linux, and macOS, was used for camera configuration and image evaluation.

Immunohistochemical staining was carried out in paraffin-embedded mouse liver specimens, following previous sectioning (2 μm thickness), dewaxing and rehydrating in graded alcohol solutions. For heat-induced epitope retrieval, the sections were heated at 100 °C for 3 × 5 min using target retrieval solution (pH 9). Non-specific binding was blocked by pre-incubation with PBS (#DM831; Dako, Glostrup, Denmark) for 20 min. Afterwards, an overnight incubation at 4 °C with the following antibodies was carried out: a. rabbit polyclonal perilipin 3/TIP47, dilution 1:100 (#ab47638; Abcam, Cambridge, United Kingdom Rabbit Polyclonal, LOT: GR3345930-9), b. rabbit polyclonal CD36, dilution 1:500 (#ab124515, Abcam Rabbit Polyclonal, LOT:GR3323919-1).

At the end of the incubation period, the slides were washed with PBS and blocked using peroxidase (#SM801, Dako) for 10 min. The slides were then incubated with EnVision Flex/HRP SM802 for 30 min. After extensive washing with PBS, the color reaction was developed in 3,3′-diaminobenzidine (DAB) for 5 min and then washed with PBS. The sections were then counterstained with hematoxylin (hematoxylin QS VECTOR Laboratories USA), dehydrated and mounted. Mean protein expression was used to group cases as having strong or weak protein expression [41]. Thus, expression of CD36 ≥ 6%, PLIN-3 ≥ 40% were defined as strong expression of each specific antibody.

### 2.7. Statistical Analysis

The statistical analysis of the collected data was carried out using IBM SPSS Statistics for Windows, Version 24.0, IBM Corp, Armonk, NY, USA. Nominal variables are presented by frequencies and valid percentages, whereas continuous ones are summarized as mean ± standard deviation (SD). Mann–Whitney U test was used for the comparison of two sample means, while Kruskal–Wallis test was used for multiple comparisons, as variables were not normally distributed. Odds ratios (OR) with their 95% confidence intervals (CI) were calculated for the nominal variables implementing the Haldane-Anscombe correction for frequencies equal to zero [35]. Statistical significance was set at *p*-value < 0.05 (two-tailed).

## 3. Results

### 3.1. NAFLD Activity Score (NAS)

A total of 95 mice were included in the study, with an average NAS of 4.96 ± 1.44. As shown in Table 1 and Figure 2, the control group had the highest NAS (5.83 ± 1.27), although the differences between groups were not statistically significant. The majority of mice in the control group developed steatohepatitis (83.33%), whereas most animals in other groups had fatty liver disease, as outlined in Table 2. The control group had significantly higher rates of steatohepatitis compared to the BASELINE group (OR: 0.01; 95% CI: 0.00–0.24; *p* = 0.005), as well as the CRO HD (OR: 0.14; 95% CI: 0.02–0.96; *p* = 0.045), CRO LD (OR: 0.10; 95% CI: 0.01–0.69; *p* = 0.020), QUE HD (OR: 0.14; 95% CI: 0.02–0.96; *p* = 0.045), QUE LD (OR: 0.10; 95% CI: 0.01–0.69; *p* = 0.020), and SLB HD (OR: 0.10; 95% CI: 0.01–0.69; *p* = 0.020) groups. The SLB LD group also had a lower rate of steatohepatitis, although the difference was not statistically significant. Representative pathology micrographs are shown in Figure 3.

### 3.2. Immunohistochemistry

The expression of CD36 and PLIN3 stratified by each group is shown in Table 3 and Table 4. Regarding CD36 expression there was a statistically significant difference among groups (*p*-value: 0.047). Further analysis revealed that experimental groups failed to reach statistically significant differences compared to the control group. Concerning PLIN3 expression, a statistically significant difference was found among groups (*p*-value: <0.001). Further analysis revealed that CRO HD (*p*-value: <0.001), QUE HD (*p*-value: 0.043), QUE LD (*p*-value: 0.002) and SLB LD (*p*-value: 0.007) exhibited a statistically significant increase in PLIN3 expression compared to the control group.

In baseline group CD36 expression was predominantly strong (63.64%), whereas in the control group the expression was evenly distributed between weak and strong expressions (Figure 4). The SLB LD group showed a significant reduction in CD36 expression compared to control, with only 8.33% showing strong expression (OR 0.09; 95% CI: 0.01–0.94; *p*-value: 0.044). Other groups did not show significant differences in CD36 expression compared to the control group.

Regarding PLIN3, the baseline group had a near-equal distribution of weak (45.45%) and strong (54.55%) expression (Figure 5). Significant increases in strong PLIN3 expression were observed in several groups. In the CRO HD group, 100% of cases showed strong expression compared to 25.00% in the control group (OR = 67.86; 95% CI: 3.12–1477.93; *p*-value: 0.007). Similarly, the QUE LD and SLB LD groups both exhibited a significant shift towards strong PLIN3 expression, with 83.33% showing strong expression in each group, compared to the control group (OR = 15.00; 95% CI: 2.02–111.18; *p*-value: 0.008 for both). These results highlight a significant upregulation of PLIN3 in these treatment groups, while other groups did not show significant differences in expression.

## 4. Discussion

Metabolic dysfunction-associated steatotic liver disease (MASLD) represents a significant global health concern, with its incidence steadily rising [34]. Although no established pharmacological treatment currently exists, various natural polyphenols—such as quercetin, silibinin, and crocetin—have demonstrated encouraging hepatoprotective effects [14]. The present study explored the hepatoprotective potential of three natural compounds, quercetin, silibinin, and crocetin, on MASLD, focusing on their effects on histopathological severity and the modulation of CD36 and PLIN3 expression, that play a pivotal role in hepatic steatosis and lipids metabolism [37,42].

The modulation of CD36 and PLIN3 observed in our study is central to the broader metabolic and inflammatory cascades driving MASLD. The downregulation of CD36 reduces fatty acid influx, thereby mitigating the lipotoxicity that triggers downstream endoplasmic reticulum (ER) stress, mitochondrial dysfunction, and pro-inflammatory NF-κB signaling [43,44]. Conversely, the upregulation of PLIN3 enhances the safe sequestration of lipids into droplets, a critical step for lipophagy, a selective form of autophagy that degrades lipids [32]. By promoting this protective autophagic flux, the compounds help clear lipid stores and alleviate cellular stress, which is often impaired in MASLD. This dual action—reducing harmful lipid uptake while improving the management and turnover of intracellular lipids—helps restore cellular homeostasis and contributes to the observed reduction in steatohepatitis, a key feature of MASH.

Quercetin, when administered in the form of its QUE-HP-β-CD lyophilized product, demonstrated substantial therapeutic efficacy in improving liver histopathology. Both low and high doses of quercetin substantially reduced the prevalence of steatohepatitis compared to the control group, highlighting its significant role in mitigating liver inflammation and injury. Steatohepatitis frequency was reduced from 83.33% in the control group to 41.67% in QUE HD and 33.33% in QUE LD. This finding is consistent with previous studies that have demonstrated quercetin’s antioxidant and anti-inflammatory effects in various models of liver injury [16,17]. Quercetin administration was associated with reduced lipid accumulation, and enhanced autophagy, findings that align closely with our observations [45]. Quercetin administration had a protective effect on NAFLD in mice by regulating intestinal microbiota imbalance and related gut-liver axis activation [21]. Furthermore, quercetin appears to act through the scavenging of reactive oxygen species (ROS), modulation of inflammatory pathways, and enhancement of mitochondrial function [46,47]. Importantly, our study demonstrated that quercetin led to significant upregulation of PLIN3, a lipid droplet-associated protein involved in safe intracellular lipid storage. Inhibiting PLIN3 may facilitate safer lipid droplet storage, limiting lipotoxicity and organelle stress within hepatocytes [32]. It is important to note the complex, context-dependent role of PLIN3. While our findings suggest a beneficial role through increased PLIN3 expression, a study by Bao et al. was protective in a model of steatohepatitis caused by hepatocyte CGI-58 deficiency in HFD-fed mice [32]. This suggests that the ultimate effect of PLIN3 may depend on the broader cellular metabolic state, where in our model, its upregulation may serve as a protective mechanism to safely sequester lipids and facilitate their turnover via lipophagy. This improved lipid droplet dynamics likely contributed to reduced hepatocellular injury and NAS. Barbier-Torres et al. showed that, while the CD36 expression was not significantly reduced in the quercetin-treated groups, the beneficial histological outcomes suggest that quercetin’s enhancement of mitochondrial fatty acid oxidation compensated for any ongoing fatty acid influx through CD36 [43]. Liu et al. suggested that quercetin inhibits CD36 expression, that was previously upregulated by HFD [44]. Therefore, quercetin likely promotes a dual mechanism: optimizing intracellular lipid turnover and stabilizing lipid droplets, thus alleviating lipotoxic stress and preventing steatohepatitis progression.

Silibinin, when administered in the form of its water soluble SLB-HP-β-CD lyophilized product, exhibited a distinct yet equally important pattern of activity. While the NAS for the silibinin low-dose and high-dose (SLB HD) groups were 5.42 ± 1.16 and 4.50 ± 1.57, respectively, only the SLB HD group showed clear improvement in steatohepatitis rates. Despite a somewhat modest reduction in NAS compared to quercetin, silibinin-treated groups exhibited a lower prevalence of steatohepatitis, indicating early molecular benefits that may precede more substantial histopathological improvements. Our findings are in line with previous studies who suggested that silibinin can inhibit hepatic lipid accumulation by modulating lipid transporter expression and enhancing lipid catabolism [48]. Serviddio and colleagues demonstrated that administration of a silybin-phospholipid complex in a murine model reduced oxidative stress and preserved mitochondrial function, thereby improving the features of non-alcoholic steatohepatitis [49]. Similarly, Elfaky et al. reported that silymarin exerted significant hepatoprotective effects in a lipopolysaccharide-induced liver injury model in Sprague Dawley rats, attenuating inflammation and liver damage [50]. Moreover, in mice with diet-induced NASH, the combination of silybin, vitamin E, and phospholipids was shown to improve insulin resistance, decrease lipotoxicity, reduce the degree of liver injury, and lower body mass index [51]. At the molecular level, silibinin treatment led to a marked upregulation of PLIN3 expression. Both the low- and high-dose silibinin groups demonstrated increased PLIN3 levels compared to controls, indicating enhanced lipid droplet stabilization. This finding aligns with the emerging understanding of PLIN3 as key modulator of lipid storage and metabolism [32]. The effect of silibinin on PLIN 3 has not been extensively discussed in the literature, but its administration has shown to regulate other PLIN family proteins, such as PLIN-2, decreasing liver lipid deposition [52]. Silibinin had a prominent effect which was the significant downregulation of CD36 expression, particularly at low doses. CD36-mediated fatty acid uptake plays a critical role in hepatic steatosis development, and its suppression is considered a promising therapeutic strategy [30]. These findings complement earlier observations that silibinin improves mitochondrial function and reduces oxidative damage [53,54]. Sugoro et al. showed an increase in CD36 expression in mice suffering from NASH, which was attenuated by silibinin and tangeretin administration by modulating CD36 towards normal levels [55]. The combined reduction in lipid influx through CD36 suppression and increased capacity for lipid storage via PLIN3 upregulation positions silibinin as a potent regulator of hepatocellular lipid homeostasis.

Crocetin treatment, when administered as lyophilized SFE, also led to marked improvements in MASLD severity. Although demonstrating moderate reductions in NAS, it had a notable impact on steatohepatitis rates showing a significant reduction in steatohepatitis prevalence from 83.33% in the control group to 33.33% and 41.67% for high-dose and low-dose groups, respectively. Crocetin’s capacity to attenuate hepatic steatosis and inflammation has been previously attributed to its antioxidant properties and mitochondrial protective effects [33,56]. Xu et al. (2021) demonstrated that crocetin mitigated hepatic inflammation and lipid accumulation in NAFLD models by enhancing antioxidant defenses and modulating the Nrf2/HO-1 pathway [56]. Similarly, Xu et reported that crocetin improved mitochondrial function and reduced oxidative stress in hepatocytes, contributing to decreased steatohepatitis severity [57]. These findings corroborate our results, suggesting that crocetin’s antioxidative and anti-inflammatory effects play a pivotal role in ameliorating steatohepatitis in NAFLD. Crocetin treatment led to the highest levels of PLIN3 expression among all groups, with all high-dosage group mice having a strong PLIN3 expression. This observation is consistent with the known properties of crocetin as a promoter of lipophagy and mitochondrial integrity [58]. By enhancing PLIN3-mediated lipid droplet stabilization and turnover, crocetin facilitates the safe handling of intracellular lipids, but available literature to strengthen these findings is limited. Crocetin did not significantly alter CD36 expression, suggesting that its therapeutic effects are independent of fatty acid uptake modulation. Zhang et al. showed that crocin, the in vivo precursor to crocetin reverses high fat induced abnormal lipid levels and lipids’ deposition by downregulating CD36 [59].

A significant strength of this study is the comprehensive approach linking histological outcomes to key lipid metabolism markers offering insight into the molecular background of histological improvements. Nonetheless, several limitations should be acknowledged. First, the use of a murine model, although well-established, may not fully replicate the complexity of human MASLD pathophysiology, including differences in lipid metabolism, immune responses, and disease progression. Furthermore, although immunohistochemistry provided valuable semi-quantitative data on CD36 and PLIN3 expressions, complementary analyses such as Western blotting or quantitative PCR could have offered more precise quantification of expression changes. Finally, our analysis did not include serum biochemical markers, such as ALT or AST, which would have provided additional evidence of hepatocellular injury. Future studies addressing these limitations will be critical for advancing these promising agents toward clinical application.

## 5. Conclusions

This experimental study demonstrates that quercetin, silibinin, and crocetin each offer distinct but complementary hepatoprotective effects against MASLD. Quercetin primarily enhanced antioxidative defense mechanisms and autophagy, leading to significant histopathological improvement and upregulation of PLIN3. Silibinin exerted a potent anti-lipogenic effect by downregulating CD36 while simultaneously regulating lipids’ metabolism through increased PLIN3 expression. Crocetin’s action also involved a marked upregulation of PLIN3, which suggests an enhanced capacity for regulating lipid storage and turnover. In conclusion, these findings underscore the critical roles of oxidative stress, autophagy, and lipid metabolism in MASLD progression and suggest that natural polyphenolic compounds targeting these pathways hold considerable promise as therapeutic agents. Further studies, including clinical trials, are warranted to validate these findings and optimize therapeutic strategies for the management of MASLD.

## Figures and Tables

**Figure 1 life-15-01523-f001:**
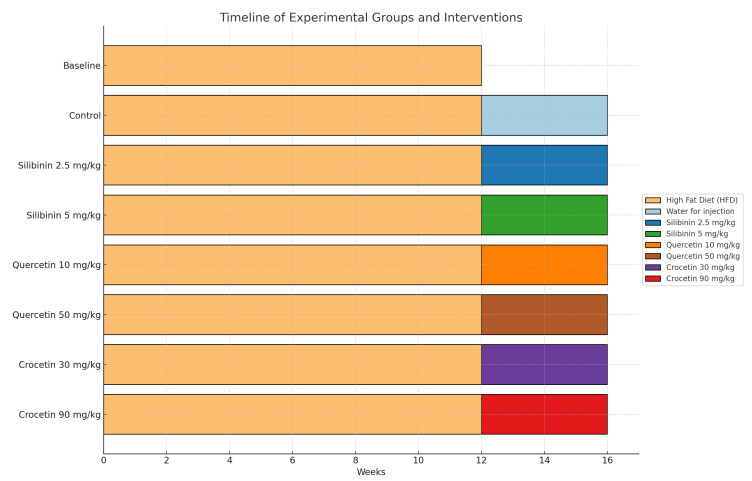
Timeline of experimental groups and interventions.

**Figure 2 life-15-01523-f002:**
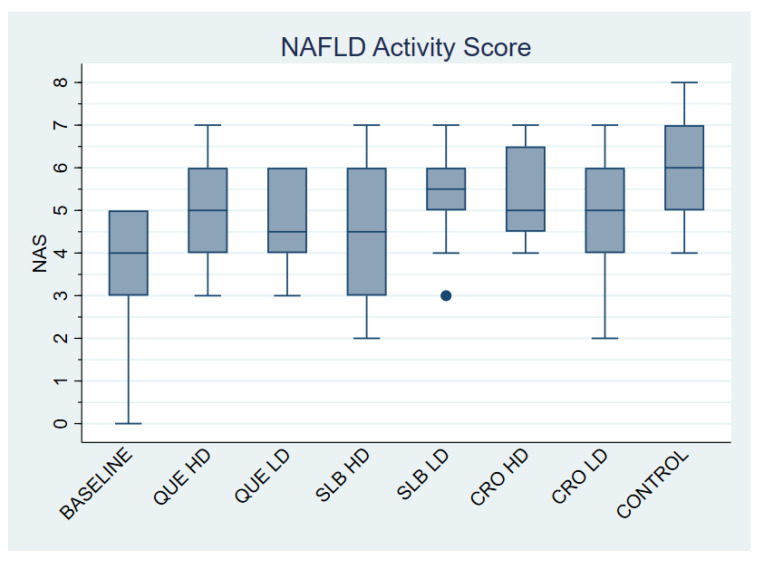
Effect of Experimental Treatments on the NAFLD Activity Score (NAS). The box plot shows the distribution of the NAFLD Activity Score (NAS) for each experimental group at the end of the study. QUE (Quercetin), SLB (Silibinin), CRO (Crocin), HD (High Dose), and LD (Low Dose).

**Figure 3 life-15-01523-f003:**
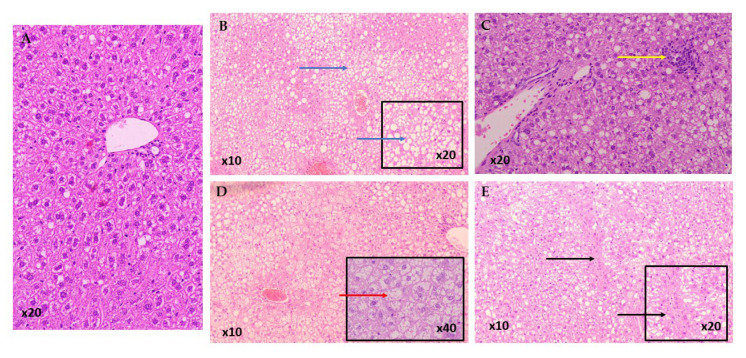
Representative Histology micrographs of liver sections showing (**A**) Normal mice liver with absence of all three hallmarks of NAFLD Activity Score, (**B**) Extensive (Grade 3) macrovesicular steatosis in >66% of hepatocytes (blue arrow), (**C**) Mild (Grade 1) lobular inflammation 1 focus per 20× field (yellow arrow), (**D**) Prominent (Grade 2) ballooning degeneration (red arrow), (**E**) Moderate (Grade 2) fibrosis (black arrow). All sections were stained with Hematoxylin and Eosin (H&E). Original magnification is shown on each image.

**Figure 4 life-15-01523-f004:**
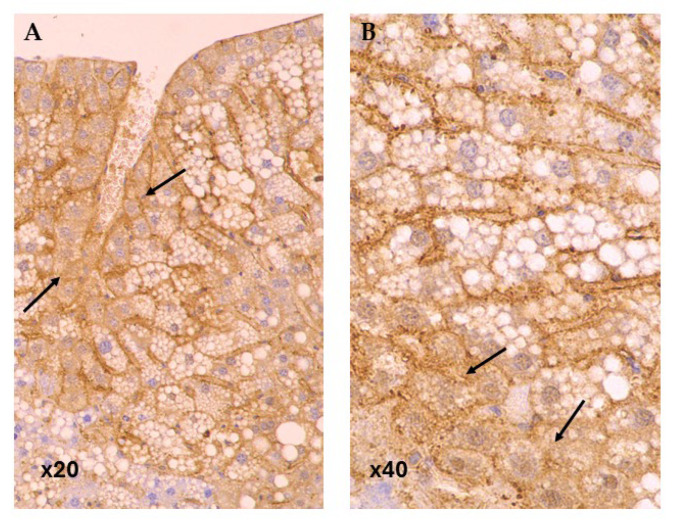
Immunohistochemical Expression of the fatty acid translocase CD36 in Steatotic Liver (black arrows). CD36 Immunostaining. Original magnification: (**A**) 20×, (**B**) 40×.

**Figure 5 life-15-01523-f005:**
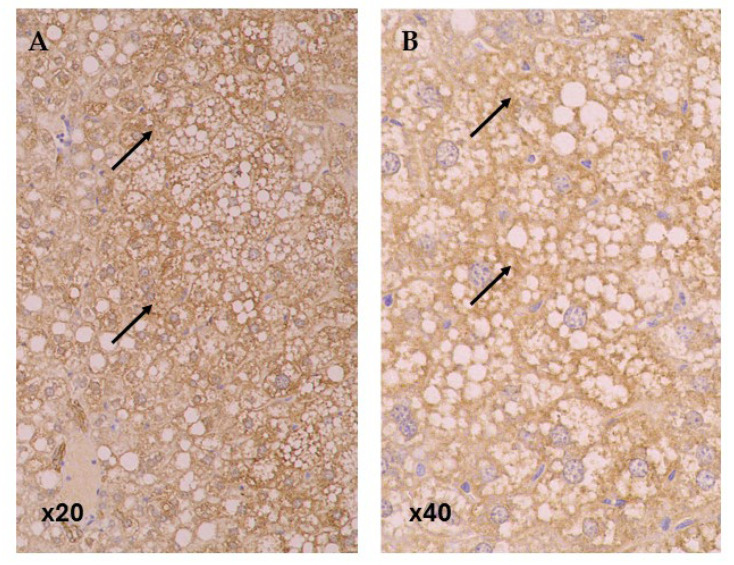
Immunohistochemical Expression of the lipid droplet-associated protein Perilipin 3 (PLIN3) in Steatotic Liver (black arrows). PLIN3 Immunostaining. Original magnification: (**A**) 20×, (**B**) 40×.

**Table 1 life-15-01523-t001:** NAFLD Activity score (NAS) stratified by each experimental group. Data are presented as mean ± standard deviation (SD).

Group	N	NAS
Baseline	11	3.73 ± 1.56
QUE HD	12	5.17 ± 1.27
QUE LD	12	4.75 ± 1.06
SLB HD	12	4.50 ± 1.57
SLB LD	12	5.42 ± 1.16
CRO HD	12	5.42 ± 1.16
CRO LD	12	4.75 ± 1.66
Control	12	5.83 ± 1.27
Total	95	4.96 ± 1.44

**Table 2 life-15-01523-t002:** Steatohepatitis prevalence stratified by each experimental group. Data are presented as a count (%).

Group	N	Fatty Liver	Steatohepatitis (NAS > 5)
Baseline	11	11 (100.00%)	0 (0.00%)
QUE HD	12	7 (58.33%)	5 (41.67%)
QUE LD	12	8 (66.67%)	4 (33.33%)
SLB HD	12	8 (66.67%)	4 (33.33%)
SLB LD	12	6 (50.00%)	6 (50.00%)
CRO HD	12	7 (58.33%)	5 (41.67%)
CRO LD	12	8 (66.67%)	4 (33.33%)
Control	12	2 (16.67%)	10 (83.33%)
Total	95	60 (63.20%)	35 (36.80%)

**Table 3 life-15-01523-t003:** CD36 and perilipin-3 (PLIN-3) expression stratified by each experimental group. Mean expression (%) is presented as mean ± standard deviation (SD).

Group	N	CD36	PLIN3
Baseline	11	7.82 ± 7.11	34.55 ± 9.34
QUE HD	12	4.00 ± 3.95	47.5 ± 23.79
QUE LD	12	8.67 ± 5.57	52.5 ± 17.65
SLB HD	12	6.83 ± 6.75	20.00 ± 8.53
SLB LD	12	2.67 ± 2.84	49.17 ± 19.29
CRO HD	12	4.42 ± 3.6	62.5 ± 11.38
CRO LD	12	4.33 ± 3.52	31.67 ± 19.46
Control	12	8.08 ± 7.17	27.50 ± 14.22
Total	95	5.83 ± 5.53	40.74 ± 20.80

**Table 4 life-15-01523-t004:** CD36 and perilipin-3 (PLIN-3) expression stratified by each experimental group. Expression strength is presented as a count (%) of animals classified as having weak or strong staining.

	CD36		PLIN3	
	weak	strong	weak	strong
Baseline	4 (36.36%)	7 (63.64%)	5 (45.45%)	6 (54.55%)
QUE HD	9 (75.00%)	3 (25.00%)	5 (41.67%)	7 (58.33%)
QUE LD	3 (25.00%)	9 (75.00%)	2 (16.67%)	10 (83.33%)
SLB HD	7 (58.33%)	5 (41.67%)	12 (100.00%)	0 (0.00%)
SLB LD	11 (91.67%)	1 (8.33%)	2 (16.67%)	10 (83.33%)
CRO HD	8 (66.67%)	4 (33.33%)	0 (0.00%)	12 (100.00%)
CRO LD	8 (66.67%)	4 (33.33%)	7 (58.33%)	5 (41.67%)
Control	6 (50.00%)	6 (50.00%)	9 (75.00%)	3 (25.00%)
Total	56 (58.95%)	39 (41.05%)	42 (44.21%)	53 (55.79%)

## Data Availability

The original contributions presented in this study are included in the article. Further inquiries can be directed to the corresponding author.

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
