# Peer review of "Therapeutic Potential of Quercetin, Silibinin, and Crocetin in a High-Fat Diet-Induced Mouse Model of MASLD: The Role of CD36 and PLIN3"

_life, 2025, doi:10.3390/life15101523_

Round 1
Reviewer 1 Report
Comments and Suggestions for Authors
Dear Authors,
The manuscript entitled "Therapeutic Potential of Quercetin, Silibinin, and Crocetin in Metabolic Dysfunction-Associated Steatotic Liver Disease (MASLD): The Role of CD36 and PLIN3 in an Experimental Study" explores the effect of three phytochemicals—quercetin, silibinin, and crocetin—on key lipid handling proteins CD36 and PLIN3 in the context of MASLD (formerly NAFLD). The study is timely, with translational relevance for the development of non-pharmacologic interventions in metabolic liver disease.
The comments that follow aim to improve the readability of the manuscript and its potential translational impact.
In the abstract, Lines 38-39 referring to "enhanced lipid handling and reduced MASLD progression through distinct mechanisms" is an overstatement. You did not investigate in depth any mechanism. Please update the abstract based on what you objectively tested. Further investigation of mechanistic pathways could be the objective of future studies.
In the Introduction section, please add more references regarding the potential beneficial effects of polyphenolic flavonoids on MASLD/ MASH (formerly NAFLD/NASH).
In the Methods Section:
- Please provide more detail on the experimental design and randomization procedures for animal group allocation.
- Please provide detailed composition of the HFD used for this study.
- The dosage justification for each compound (quercetin, silibinin, crocetin) should be supported with references or pilot data. Each compound is administered in different amounts (mg/kg). What is the potential of each compound? Would these differences impact the results?
- Please clarify whether power calculations were performed for the n of animals included in the study and each group.
- Please clearly describe the histological scoring criteria and whether it was performed blinded. The manuscript would benefit from the addition of images showing different NAS grades and respective histological features.
- How were the cutoffs for CD36 and PLIN3 defined? Please provide more information on the how expression levels were assessed and the whether the cutoffs of expression are arbitrary or previously established.
In the Results section:
- The manuscript would benefit from the addition of graphs and possibly showing localization of CD36 and PLIN3 expression levels in the hepatic sections.
- Please clarify whether the observed effects were primarily preventative or therapeutic (i.e., administered before or after steatosis onset).
- Some claims about “protective” effects could be better supported by including ALT/AST data or inflammatory markers.
In the Discussion section:
- Line 281: "...in HFD-fed". Complete the sentence.
- Lines 284-287: You have not directly tested the mechanism of mitochondrial fatty acid oxidation. Your suggestion is an overstatement. Please rephrase to clearly state what can be safely concluded from the present study vs. what has been found in the literature from other studies.
- The discussion should expand on how modulation of CD36 and PLIN3 by the phytochemicals relates to broader metabolic and inflammatory pathways in MASLD.
In the References section, please check consistency of referencing. For instance, references 14 and 27 lack the name of the journal. Please update.
Sincerely yours.
Comments on the Quality of English LanguageImprove sentence structure and grammar throughout the manuscript. For instance, rephrase long or unclear sentences, and ensure subject-verb agreement.
Author Response
Comment 1: Dear Authors,
The manuscript entitled "Therapeutic Potential of Quercetin, Silibinin, and Crocetin in Metabolic Dysfunction-Associated Steatotic Liver Disease (MASLD): The Role of CD36 and PLIN3 in an Experimental Study" explores the effect of three phytochemicals—quercetin, silibinin, and crocetin—on key lipid handling proteins CD36 and PLIN3 in the context of MASLD (formerly NAFLD). The study is timely, with translational relevance for the development of non-pharmacologic interventions in metabolic liver disease.
The comments that follow aim to improve the readability of the manuscript and its potential translational impact.
In the abstract, Lines 38-39 referring to "enhanced lipid handling and reduced MASLD progression through distinct mechanisms" is an overstatement. You did not investigate in depth any mechanism. Please update the abstract based on what you objectively tested. Further investigation of mechanistic pathways could be the objective of future studies.
Response 1: Thank you for your valuable input. Abstract was modified to better reflect the study.
Comment 2: In the Introduction section, please add more references regarding the potential beneficial effects of polyphenolic flavonoids on MASLD/ MASH (formerly NAFLD/NASH).
Response 2: We appreciate this suggestion. We have enriched the Introduction section with several additional and more recent references that discuss the hepatoprotective effects of various natural polyphenols in the context of MASLD, providing a broader background for our study.
Comment 3:
In the Methods Section:
- Please provide more detail on the experimental design and randomization procedures for animal group allocation.
Response 3: We updated the relevant Methods section and also cite the established protocol of our facility for this experimental study.
Comment 4:
- Please provide detailed composition of the HFD used for this study.
Response 4: We have updated the Methods section to include the detailed composition of the high-fat diet.
Comment 5:
- The dosage justification for each compound (quercetin, silibinin, crocetin) should be supported with references or pilot data. Each compound is administered in different amounts (mg/kg). What is the potential of each compound? Would these differences impact the results?
Response 5: Thank you for your comment. In the revised manuscript (Section 2.1), we have added the relevant references to our previous work as well as other studies supporting the dose selection for each compound. Since the compounds have different physicochemical and pharmacological properties, they are expected to be administered at different doses. We chose to apply the same doses as in our previous related studies to ensure consistency and comparability, and because we have already assessed the general impact (e.g., toxicity) of these doses.
Comment 6:
- Please clarify whether power calculations were performed for the n of animals included in the study and each group.
Response 6: An a priori power analysis was conducted using G*Power to determine the necessary sample size, as required for bioethical approval. With parameters set to a power (1−β) of 0.8, a significance level (α) of 0.05, and a large effect size (f = 0.4), the analysis indicated a target of 104 mice (n=13 per group). However, due to limited animal availability, the experiment proceeded with a final total of 95 mice.
Comment 7:
- Please clearly describe the histological scoring criteria and whether it was performed blinded. The manuscript would benefit from the addition of images showing different NAS grades and respective histological features.
Response 7: Histological analysis was performed by an expert pathologist (A.G.) who was blinded to the experimental groups. Non-alcoholic fatty liver disease (NAFLD) was evaluated using the NAFLD Activity Score. Furthermore, we have included representative histological images in the Results section.
Comment 8:
- How were the cutoffs for CD36 and PLIN3 defined? Please provide more information on the how expression levels were assessed and the whether the cutoffs of expression are arbitrary or previously established.
Response 8: Thank you for your comment. As described previously in the literature, we used mean values of CD36 and PLIN3 as cut-offs for the level of expression.
Comment 9:
In the Results section:
- The manuscript would benefit from the addition of graphs and possibly showing localization of CD36 and PLIN3 expression levels in the hepatic sections.
Response 9: Thank you for your comment. Representative images were added in the Results section.
Comment 10:
- Please clarify whether the observed effects were primarily preventative or therapeutic (i.e., administered before or after steatosis onset).
Response 10: Thank you for your comment. We have clarified in the Methods section that the interventions were therapeutic. The compounds were administered for 4 weeks after hepatic steatosis was first established by feeding the mice a high-fat diet for 12 weeks.
Comment 11:
- Some claims about “protective” effects could be better supported by including ALT/AST data or inflammatory markers.
Response 11: We agree that including biochemical markers such as ALT/AST or cytokine levels would have strengthened our findings. Unfortunately, these analyses were not performed in the current study. We have acknowledged this as a limitation in the Discussion section and suggest that future studies should incorporate these valuable endpoints.
Comment 12:
In the Discussion section:
- Line 281: "...in HFD-fed". Complete the sentence.
Response 12: Thank you for your comment. This section was rephrased.
Comment 13:
- Lines 284-287: You have not directly tested the mechanism of mitochondrial fatty acid oxidation. Your suggestion is an overstatement. Please rephrase to clearly state what can be safely concluded from the present study vs. what has been found in the literature from other studies.
Response 13: We have revised the Discussion section to clearly distinguish our study's findings from the mechanisms proposed in the cited literature.
Comment 14:
- The discussion should expand on how modulation of CD36 and PLIN3 by the phytochemicals relates to broader metabolic and inflammatory pathways in MASLD.
Response 14: We have expanded the Discussion to better contextualize our findings. We now elaborate on how the observed changes in CD36 and PLIN3 likely intersect with key pathways in MASLD, such as insulin resistance, oxidative stress, and inflammation.
Comment 15:
In the References section, please check consistency of referencing. For instance, references 14 and 27 lack the name of the journal. Please update.
Sincerely yours.
Response 15: Thank you for your comment. We used EndNote reference manager using the MDPI style.
Comment 16:
Comments on the Quality of English Language
Improve sentence structure and grammar throughout the manuscript. For instance, rephrase long or unclear sentences, and ensure subject-verb agreement.
Response 16: A comprehensive proofread of the entire manuscript was performed by a native speaker in order to improve grammar, clarity, and sentence structure, addressing sentences pointed out by the reviewers.
Reviewer 2 Report
Comments and Suggestions for Authors
life-3776483
In this paper, the authors focus on the potential role of the bioactive compounds quercetin, crocetin, and silibinin in CD36 and Plin3 in MAFLD. Unfortunately, this work is poorly detailed and lacks new information, and it lacks relevant methodological elements.
In the title the authors should to identify the model, also they should improve keywords, it supposes not to repeat the same words as title
In introduction…authophagy????
Due there are some studies on liver damage models, the novelty is not clear
The introduction is to long, please shortened.
In the experimental model, the model definition is poorly written (tangled); furthermore, the authors must report the number of animals per experimental group and the fat content of the high-fat diet (and the brand with its respective calories).95 animals is a very large number considering the 3Rs. This is unethical. I request that you attach the complete document approved by the animal ethics committees that approved this project.
There is no control of quercetin, silibinin or crocetin (a control of doses without HFD), otherwise it is not possible to know if the compounds themselves are safe for the liver (not enough information by literature to ensure that they are safe).
The authors report the use of Stigmas of Crocus sativus L. (saffron), but they do not indicate how they ensure purity and that there are no other interferences in the sample.
The NAS score is not the most appropriate for this type of study. Authors should perform a Korourian-type semiquantitative analysis. They should also include microphotographs with all the findings.
Immunohistochemical techniques are qualitative, so under no circumstances can they only be presented in numerical terms (semi-quantitative), otherwise it is not credible. Likewise, they should explain the data processing behind these tables.
the discussion must be completely redone.
at least 10% of the references are self-citations
Author Response
Comment 1:
In this paper, the authors focus on the potential role of the bioactive compounds quercetin, crocetin, and silibinin in CD36 and Plin3 in MAFLD. Unfortunately, this work is poorly detailed and lacks new information, and it lacks relevant methodological elements.
Response 1: Thank you for your valuable input. We have taken the reviewers’ feedback seriously and have substantially expanded the Materials and Methods section with more details as requested.
Comment 2:
In the title the authors should to identify the model, also they should improve keywords, it supposes not to repeat the same words as title
Response 2: Thank you for your comment. Title modified to clearly identify the model and keywords were updated.
Comment 3:
In introduction…authophagy????
Response 3: We have revised the introduction to better explain the critical role of autophagy in hepatic lipid metabolism and how its dysregulation contributes to MASLD, thereby establishing a clear rationale for investigating the autophagy-related proteins CD36 and PLIN3.
Comment 4:
Due there are some studies on liver damage models, the novelty is not clear
Response 4: Thank you for your comment. While these compounds have been studied individually, our study is novel in its simultaneous comparison of quercetin, silibinin, and crocetin within the same experimental model of MASLD. Furthermore, our focus on their differential effects on the key lipid-regulating proteins CD36 and PLIN3 provides new insights into their distinct potential mechanisms.
Comment 5:
The introduction is to long, please shortened.
Response 5: Thank you for your comment. We have revised the introduction to better address reviewers’ comments.
Comment 6:
In the experimental model, the model definition is poorly written (tangled); furthermore, the authors must report the number of animals per experimental group and the fat content of the high-fat diet (and the brand with its respective calories).95 animals is a very large number considering the 3Rs. This is unethical. I request that you attach the complete document approved by the animal ethics committees that approved this project.
Comment 6: An a priori power analysis was conducted using G*Power to determine the necessary sample size, as required for bioethical approval. With parameters set to a power (1−β) of 0.8, a significance level (α) of 0.05, and a large effect size (f = 0.4), the analysis indicated a target of 104 mice (n=13 per group). However, due to limited animal availability, the experiment proceeded with a final total of 95 mice.
An ethical approval was obtained by the Veterinary Authorities of Region of Athens, Greece (Ethical approval num. no. 792333/4.12.2019) and the Bioethics Committee of National and Kapodistrian University of Athens (no. 239/28.1.2020)
Comment 7:
There is no control of quercetin, silibinin or crocetin (a control of doses without HFD), otherwise it is not possible to know if the compounds themselves are safe for the liver (not enough information by literature to ensure that they are safe).
Response 7:
Thank you for your comment. Quercetin and silibinin are insoluble in water and therefore cannot be administered in water for injection (WFI) as stand-alone controls. In the case of saffron, it is not pure crocetin that is administered, but a standardized lyophilized aqueous saffron extract, from which the content of the main constituent, crocin, is quantified and expressed as equivalent of crocetin (using the MW of both substances), to determine the theoretical administered dose of crocetin.
Regarding safety, available evidence supports the hepatic safety—and in some cases, hepatoprotective properties—of the tested compounds at the administered doses:
Quercetin: It is a well-established natural antioxidant with anti-inflammatory and hepatoprotective properties. Also, recently a study demonstrated its protective role against metabolic-associated fatty liver disease by modulating hepatic lipid metabolism, oxidative stress, and inflammation (Ling Jiang et al., 2025, Animal Biotechnology, 36:1, 2442351, DOI: 10.1080/10495398.2024.2442351).
Silibinin: It is well established as a hepatoprotective agent with antioxidant, anti-inflammatory, and anti-fibrotic properties. Multiple in vivo studies, including our own, have confirmed its protective effects on the liver following ischemia/reperfusion injury, both histologically and via downregulation of key pro-inflammatory mediators such as TNF-α, IL-6, and MCP-1 (Kyriakopoulos G et al., 2022, Basic Clin. Pharmacol. Toxicol. DOI:10.1111/bcpt.13704; Betsou A et al., 2021, J Pharm Pharmacol. DOI:10.1093/jpp/rgab062).
Saffron extract: Considered safe for the liver at comparable doses, as documented in a study on non-alcoholic fatty liver disease in diabetic rats (Konstantopoulos P et al., 2017. Biomed Rep. DOI: 10.3892/br.2017.884), as well as in a randomized, double-blind, placebo-controlled study on patients with non-alcoholic fatty liver disease (Reyhane Sadat Mirnasrollahi Parsa et al., 2024, Herb. Med. DOI: 10.1016/j.hermed.2024.100877).
Based on these considerations, we are confident that the administered doses of these compounds are safe for the liver under the study conditions.
Comment 8:
The authors report the use of Stigmas of Crocus sativus L. (saffron), but they do not indicate how they ensure purity and that there are no other interferences in the sample.
Response 8:
Thank you for your comment. The saffron aqueous extract used in our study was prepared and chemically characterized according to a validated procedure described in Christodoulou et al. (2019, J Pharm Pharmacol, 71, 753–764, doi:10.1111/jphp.13055). The stigmas of Crocus sativus L. were sourced from the Kozani Saffron Producers Cooperative (Kozani, Greece), and a voucher specimen was deposited at the Herbarium of the Laboratory of Pharmacognosy and Chemistry of Natural Products, NKUA. The extract (SFE) was prepared under controlled conditions (1 g stigmas in 37.5 ml HPLC water, refrigerated for 3 days, filtered under low pressure, and freeze-dried) and analyzed by semi-preparative HPLC-PDA and ¹H- and 2D-NMR to identify and quantify its main constituents. The chemical profile revealed that SFE consisted mainly of all-trans-crocin, picrocrocin, and picrocrocin aglycon, with very low safranal content, consistent with the aqueous nature of the extract. All peaks were identified by retention time comparison with authentic standards and confirmed by NMR spectra, ensuring the absence of unknown interfering components in significant amounts. Additionally, quantification of crocin content was performed by a validated HPLC-PDA method. Therefore, the preparation, voucher authentication, chromatographic separation, spectroscopic confirmation, and quantitative assay ensure the purity of the extract and exclude the presence of major interfering substances that could confound biological effects in our study.
A relevant sentence has been added at the end of section 2.3
Comment 9:
The NAS score is not the most appropriate for this type of study. Authors should perform a Korourian-type semiquantitative analysis. They should also include microphotographs with all the findings.
Response 9: Thank you for this suggestion. We acknowledge that alternative scoring systems like the Korourian-type analysis provide valuable data. For this study, we chose the NAFLD Activity Score (NAS) because it is the most widely validated and accepted histological scoring system for both preclinical and clinical MASLD studies, as established by the NASH Clinical Research Network, ensuring our results are highly comparable to the broader literature. We believe it remains the most appropriate standard for our primary histological outcome. We have added representative images of the IHC staining, allowing for a visual assessment of the protein expression and localization
Comment 10:
Immunohistochemical techniques are qualitative, so under no circumstances can they only be presented in numerical terms (semi-quantitative), otherwise it is not credible. Likewise, they should explain the data processing behind these tables.
Response 10: Thank you for your comment. NAFLD activity score and proteins expression were assessed and a statistical analysis was performed based on these results.
Comment 11:
the discussion must be completely redone.
Response 11: In response to the detailed feedback from all three reviewers, we have substantially revised the Discussion section. It is now more balanced, distinguishes our findings from the literature more clearly, avoids overstatement, discusses the study's limitations, and better contextualizes the roles of CD36 and PLIN3.
Comment 12:
at least 10% of the references are self-citations
Response 12: Thank you for your feedback. Three of the 60 total references are from our immediate research group; these include two literature reviews that establish the broader context for this work and one related experimental study. Furthermore, several references in the Methods section are from collaborating groups within our institution. These citations are methodologically essential, as they describe the specific, validated procedures for compound preparation and dose selection used in the current study, ensuring transparency and reproducibility. Therefore, we believe the inclusion of these references is appropriate and justified.
Reviewer 3 Report
Comments and Suggestions for Authors
See the attached file

English and grammar must be revised.
Author Response
Comment 1:
The article entitled “Therapeutic Potential of Quercetin, Silibinin, and Crocetin in Metabolic Dysfunction-Associated Steatotic Liver Disease (MASLD): The Role of CD36 and PLIN3 in an Experimental Study” addresses a clinically relevant and increasingly urgent biomedical issue which is the development of safe, natural, and effective therapeutic modality for Metabolic Dysfunction-Associated Steatotic Liver Disease (MASLD), a major contributor to global morbidity with limited pharmacologic interventions currently approved. The investigation into polyphenolic compounds such as quercetin, silibinin, and crocetin, and their modulation of autophagy-related lipid regulators CD36 and PLIN3, is both innovative and timely, particularly in light of recent insights into lipid droplet biology and lipophagy.
It demonstrated that quercetin, silibinin, and crocetin each offer different yet complementary hepatoprotective effects against MASLD. Quercetin primarily improved liver histopathology and increased PLIN3 expression. Silibinin significantly reduced steatohepatitis, upregulated PLIN3, and notably downregulated CD36. Crocetin markedly improved disease severity and showed the highest PLIN3 expression. These findings highlight the potential of these natural compounds to enhance lipid handling and reduce MASLD progression by modulating autophagy and lipid metabolism, identifying PLIN3 and CD36 as promising therapeutic targets.
However, the manuscript needs English proofreading and grammar revision, as well as major issues that I will discuss below:
Title: The title is informative but lengthy; consider rephrasing. An experimental study is a wide term; consider specifying the model name, like HFD mice.
Response 1: Thank you for your comment. We have revised the title to be more concise.
Comment 2:
Abstract:
The background address MASLD is a pressing metabolic disorder lacking effective therapy. However, it could briefly mention its global prevalence and recent redefinition. Nothing was mentioned regarding the used compounds.
Line 26-27: could be rephrased because much research is going to develop natural or synthetic therapeutics.
Line 27–28: "This study investigates the therapeutic potential of natural compounds, quercetin, silibinin, and crocetin, in a experimental model of MASLD."
Grammatical error: “a experimental” should be “an experimental”.
The groups should be clearly described.
Line 33-34: expression of PLIN3 and CD36, you should clarify that it is protein expression.
In the results summary, you would better add the p-value.
Response 2: Thank you for your input. We critically revised the abstract to reflect your suggestions.
Comment 3:
Introduction:
The authors reviewed well the pathogenesis of MASLD, nomenclature, theory, and epidemiology.
However, here are some comments: -
Mention some about the background of the used drugs.
Use transitional phrases among paragraphs.
Add more about the PLIN3 and CD36 gene expressions and argue why the authors used those genes to draw the hypothesis, and what other genes are involved in those pathways.
Response 3: Thank you for your comment. Introduction was rephrased to address your comments.
Comment 4:
Add also paragraph about clinical trials used with those compounds, if any, and use up-to- date literature and avoid outdated ones.
Response 4: Thank you for your comment. We added relevant results from clinical trials.
Comment 5:
Avoid self-citation and plagiarism, and do it all over the manuscript.
Response 5: Thank you for your feedback. Three of the 60 total references are from our immediate research group; these include two literature reviews that establish the broader context for this work and one related experimental study. Furthermore, several references in the Methods section are from collaborating groups within our institution. These citations are methodologically essential, as they describe the specific, validated procedures for compound preparation and dose selection used in the current study, ensuring transparency and reproducibility. Therefore, we believe the inclusion of these references is appropriate and justified.
Comment 6:
Line 117–120: “This study aimed to assess the effect of natural compounds (quercetin and silibinin when administered in the form of their lyophilized products with hydroxypropyl-β- cyclodextrin QUE-HP-βCD and SLB-HP-β-CD, as well as crocetin when administered in the form of its lyophilized aqueous saffron extract SFE) on MASLD..."
This sentence is long and complex. It would benefit from restructuring or splitting to be easy for the reader.
Response 6:
Thank you for your suggestion. The sentence has been restructured in the revised manuscript.
Comment 7:
Materials and Methods
Although written with details, here are some major concerns
The age and weight of animals, housing condition, no of animals per cage, humidity, temperature, acclimatization period, chow component, and HFD components, and if you intend to add a reference, you should mention as previously described or according to,
....... etc.
Response 7: We have expanded Section 2.1 to include all the requested methodological details: animal age and weight at the start of the study, housing conditions (cages per animal, temperature, humidity, light/dark cycle), acclimatization period, and details of the chow and HFD.
Comment 8:
Doses must be revised and justify why those doses were selected by reference.
Doses and preparation, molarity and conversion should be revised and rewritten in a clear format to help reproducibility or supported by references of earlier research, and avoid self- plagiarism as much as possible. Ensure consistent use of units (e.g., mg/kg vs. g/kg).
Response 8:
Thank you for your comment. In the revised manuscript (Section 2.1), we have added the relevant references to our previous work as well as other studies supporting the dose selection for each compound. Additionally, the equivalent of each dose in milligrams is included to allow for reproducibility.
Comment 9:
Graphical representation of the experimental design and symbols for long chemical names of the used therapeutics could make it easier for the reader to follow.
Response 9: We added a Figure summarizing experimental design and control groups.
Comment 10
Line 134–145: Dose description for the silibinin groups.
There is a likely typing error in the silibinin dosing. The low-dose group is listed as “silibinin 2.5g/kg” (line 139), while the high-dose is “5mg/kg” (line 141). This appears inconsistent 2.5 g/kg is 2500 mg/kg, which is an extremely high dose compared to 5 mg/kg.
Please verify and correct the doses. Based on typical silibinin studies and the context of other compounds, it is likely that the low dose should be 2.5 mg/kg, not 2.5 g/kg. If 2.5 g/kg was indeed used, this should be justified with a reference, as it is pharmacologically unusual
Response 10:
Thank you for pointing this out; it was indeed a typographical error. The doses have been corrected in the revised manuscript and are now also presented in milligrams.
Comment 11
Line 169–170: The QUE content in the lyophilized product was determined to be 7.3 ± 0.15% (w/w), while the SLB content was 11.11 ± 0.68% (w/w). These percentages are critical for doses accuracy. Please clarify whether these values were used to adjust the administered volumes to ensure precise quercetin and silibinin dosing (e.g., 10 mg/kg quercetin, not QUE-HP-βCD). Please revise and write the dosages part in the methodology accurately to ensure reproducibility.
Response 11
Thank you for your comment. We confirm that the reported percentages of quercetin (7.3 ± 0.15% w/w) and silibinin (11.11 ± 0.68% w/w) in the lyophilized products were indeed used to adjust the administered volumes and ensure accurate dosing of the active compounds. All doses reported in the manuscript (e.g., 10 mg/kg for quercetin and 50 mg/kg for silibinin) refer to the amount of pure active compound (e.g. quercetin), not the total weight of the inclusion complex or lyophilized product. The doses are also reported in milligrams of the active compound
.
Comment 12:
Additionally, line 177–178: “The extract contained 27.8 (±0.1) % w/w of crocin (equivalent to 9.34% w/w of crocetin...” The conversion from crocin to crocetin is important. Please clarify the stoichiometric basis for this conversion (e.g., molecular weight ratio). Also, confirm that the administered crocetin doses (2.84 mg/kg and 8.51 mg/kg) are based on this hydrolysis equivalence. Discuss the bioavailability of both drugs since crocetin is lipid-soluble and the other is water-soluble.
Response 12:
Thank you for your suggestion. The conversion has been clarified in the revised manuscript (section 2.1). We would like to note that the administered material was not pure crocetin, but a standardized lyophilized aqueous saffron extract (SFE). The content of its main constituent, crocin, was quantified and expressed as crocetin equivalent (based on the molecular weights of both compounds) to determine the theoretical administered dose of crocetin [28]. The bioavailability of both compounds has been previously investigated (Christodoulou et al., 2019. J Pharm Pharmacol. doi: 10.1111/jphp.13055). Notably, crocin levels in blood are undetectable, as SFE-derived crocin is rapidly hydrolyzed in the gastrointestinal tract to crocetin (Asai et al., 2005. J Agric Food Chem. doi: 10.1021/jf0509355).
Comment 13:
Basic immunohistochemistry protocol is provided (Lines 187–202), but key methodological elements are missing, such as Antibody validation (specificity, source, lot numbers), negative controls, blinding in image scoring, and reference or justification for thresholds (CD36 ≥6%, PLIN3 ≥40%).
Moreover, methods lack details regarding the immunohistochemical (IHC) analysis of CD36 and PLIN3 protein expression, as well as the associated imaging techniques and statistical transparency in data presentation.
Clearly describe the scoring system used for CD36 or PLIN3, and state whether quantification was manual, semi-quantitative, or digitally performed.
There is no mention of the criteria applied to define “strong” versus “weak” staining, nor are any references provided to justify the positivity thresholds (e.g., ≥6% for CD36, ≥40% for PLIN3).
The scoring methodology is missing.
Microscopy and imaging systems used. The manuscript omits key details such as the microscope model, objective magnification, camera type, image acquisition software, and any post-processing steps—information necessary for reproducibility and for readers to assess the validity of visual analyses.
Response 13: Thank you for your comment. Immunohistochemistry analysis was performed blindly by an expert pathologist. Antibody validation details were added in the Methods section. The pylon Viewer was used for camera configuration and image evaluation. As described previously in the literature, we used mean values of CD36 and PLIN3 as cut-offs for the level of expression.
Comment 14:
Statistical Analysis
Line 204–212: Appropriate tests are listed (t-test, Mann-Whitney U), but there is no correction for multiple comparisons.
Response 14: Thank you for your comment. We compared each group to the control group. Post hoc analysis was performed for multiple comparisons.
Comment 15:
Line 230–233: SLB-LD shows p = 0.040—this may lose significance after Bonferroni or Tukey correction, add post-hoc test.
Response 15: This was a Mann-Whitney U test and so no post-hoc test was performed.
Comment 16:
Using T-tests and Mann-Whitney U tests is suboptimal for analyzing 8 groups experimental design. I recommend that the authors apply One-Way ANOVA (with post-hoc Tukey’s HSD) for normally distributed data or the Kruskal-Wallis test (with Dunn’s post-hoc) for non- parametric outcomes. Importantly, since the experimental design includes both compound identity and dose, the use of Two-Way ANOVA or its non-parametric analog, such as Aligned Rank Transform (ART) ANOVA, would allow the authors to test for both main effects and interaction effects between treatment and dose. This approach is statistically more powerful and biologically more informative than isolated pairwise comparisons. Additionally, the manuscript should clearly state whether data were assessed for normality and homogeneity of variances, and all p-values in tables should be linked to the exact statistical test applied.
Response 16: Thank you for your comment. We compared each group to control group so only two samples test were applied. Kruskal Wallis test with post hoc analysis was used in multiple comparison tests.
Comment 17:
Results
The results are poorly presented and lack a critical part of the manuscript H&E photomicrographs to prove the occurrence of MASLD with the characteristic histopathological criteria and morphology are missing.
Immunohistochemistry photomicrographs and scoring criteria, characterization of the expressed proteins, and their subcellular localization before and after therapy.
Response 17: Thank you for your input. We have added three figures with representative IHC images.
Comment 18:
Tables are lacking the detailed key of the statistical test used, significance, etc.
Response 18: Thank you for your comment. Tables present the results of our study. Each group was compared to the control group and, thus, no such data was presented at the tables.
Comment 19:
Line 215–216: The control group had the highest NAS score (5.83 ± 1.27), although the differences between groups were not statistically significant.
This statement is misleading. It would be better to present data in histograms or incorporate p-values in the tables.
Response 19: Thank you for your comment. We added respective boxplots for NAS score of each group.
Comment 20:
Line 220–223: Reporting of odds ratios (ORs) for steatohepatitis. The ORs are very low (e.g., 0.01 for baseline vs. control), which may reflect the near-absence of steatohepatitis in the baseline group. However, the 95% CI for the baseline group includes zero (0.00–0.24), which is challenging. Clarify this.
Response 20: Thank you for your input. Haldane-Anscombe correction for frequencies equal to zero was applied.
Comment 21
Line 230–233: Regarding CD36 expression, there was a statistically significant difference among groups (p-value: 0.047)... SLB LD group demonstrated a significantly lower expression (p-value: 0.040)"
The use of multiple pairwise comparisons without correction for multiple testing (e.g., Bonferroni or Tukey) increases the risk of Type I error. Given that seven comparisons were likely made (vs. control), the threshold for significance should be adjusted. The p = 0.040 may no longer be significant after correction. Apply a post-hoc test.
Response 21: Thank you for your comment. After applying a post-hoc test the SLB LD group had no statistical significant difference to the control group and we corrected it in the results section.
Comment 22:
Overall, the statistical presentation within Tables 2, 3, and 4 lacks sufficient clarity. The tables legends do not specify which statistical tests were used for each comparison, and no corrections for multiple testing are described, despite the use of numerous group comparisons. P-values are inconsistently reported, and it is not clear whether they are derived from parametric or non-parametric analyses. In addition, Tables 3 and 4 present similar data in different formats (continuous vs. categorical) without clarifying their distinct purposes, creating confusion, and would be better combined.
Response 22: Thank you for your comment. We modified the table legends to better reflect their contents. We compared each group to the control so no comparisons are presented at the Tables. Table 3 presented the expression of PLIN3 and CD36, while Table 4 presented the strong vs weak expression of each group.
Comment 23:
Discussion
Line 278–282:
The authors state that "Inhibiting PLIN3 may facilitate safer lipid droplet storage" (line 281– 282), citing Bao et al. However, Bao et al. showed that knockdown of PLIN3 prevented steatosis, suggesting that PLIN3 promotes steatosis. This contradicts the interpretation that increased PLIN3 (as seen with quercetin) is beneficial.
Discuss these opposing results from other groups. Mostly, PLIN3 upregulation in the context of enhanced autophagy (as induced by quercetin) promotes a safe storage and turnover of lipids rather than steatosis.
Response 23: Thank you for this insightful comment. We agree the role of PLIN3 is context-dependent. The cited study involved a specific genetic defect in lipolysis . In our HFD model, we propose PLIN3 upregulation is a protective response to enhance safe lipid storage and turnover. We revised the discussion accordingly.
Comment 24:
Lines 343–344: The authors stated that Zhang et al. showed that crocetin reverses high-fat- induced abnormal lipid levels and lipid deposition by downregulating CD36 [54].
However, reference [54] discusses crocin, not crocetin. Mention the difference between or add the word precursor of ...
Response 24: Thank you for this correction. We have amended the sentence in the discussion to clarify that the cited study investigated crocin, which is the in-vivo precursor to crocetin, the molecule of interest in our study.
Comment 25:
Conclusion
line 363–364: “The conclusion that crocetin 'facilitates lipid storage regulation and promotion of lipid turnover” is probable but not directly supported by the data, as no autophagy markers (e.g., LC3, p62) were measured for crocetin.
Response 25: We have revised the Conclusion to be more cautious, stating that the significant upregulation of PLIN3 by crocetin suggests an enhanced capacity for lipid droplet handling, rather than stating it as a proven fact from our study.
Comment 26:
References:
Try to cite more recent 2025 work and avoid old references (2000) if they can be replaced by recent literature.
Avoid self-citation
Follow the journal format for reference style, make all uniform.
Response 26: Thank you for your comment. We used EndNote Reference Manager using MDPI style for the references of this manuscript. The vast majority of citations are recent (published after 2015). The few older references included are foundational papers that provide essential historical context for the work (eg 10.1016/s0016-5085(98)70599-2). Three of the 60 total references are from our immediate research group; these include two literature reviews that establish the broader context for this work and one related experimental study. Furthermore, several references in the Methods section are from collaborating groups within our institution. These citations are methodologically essential, as they describe the specific, validated procedures for compound preparation and dose selection used in the current study, ensuring transparency and reproducibility. Therefore, we believe the inclusion of these references is appropriate and justified.
Round 2
Reviewer 1 Report
Comments and Suggestions for Authors
Dear Authors,
The revised manuscript entitled “Therapeutic Potential of Quercetin, Silibinin, and Crocetin in a High-Fat Diet-Induced Mouse Model of MASLD: The Role of CD36 and PLIN3” is significantly improved compared to the previous version. The introduction has been expanded to include updated nomenclature (MASLD/MASH), current guidelines, and mechanistic context. Methods are better detailed, statistical analyses clarified, and figures/tables improved. Overall, the paper is scientifically sound, relevant, and contributes meaningfully to the field of MASLD therapeutics.
To further strengthen the work, I recommend improving clarity in figures.
For Figures 3 & 4, please consider adding scale bars that are clearly visible.
For Figure 5, the PLIN3/CD36 staining images could benefit from higher magnification insets to highlight cellular localization.
Sincerely yours.
Author Response
Comment 1: Dear Authors,
The revised manuscript entitled “Therapeutic Potential of Quercetin, Silibinin, and Crocetin in a High-Fat Diet-Induced Mouse Model of MASLD: The Role of CD36 and PLIN3” is significantly improved compared to the previous version. The introduction has been expanded to include updated nomenclature (MASLD/MASH), current guidelines, and mechanistic context. Methods are better detailed, statistical analyses clarified, and figures/tables improved. Overall, the paper is scientifically sound, relevant, and contributes meaningfully to the field of MASLD therapeutics.
To further strengthen the work, I recommend improving clarity in figures.
For Figures 3 & 4, please consider adding scale bars that are clearly visible.
For Figure 5, the PLIN3/CD36 staining images could benefit from higher magnification insets to highlight cellular localization.
Sincerely yours.
Response 1: Thank you for your positive feedback and constructive suggestions to improve the manuscript. We have revised the figures to enhance their clarity. We included magnification scales on each image. Furthermore, higher magnification for PLIN3 and CD36 images were added to highlight cellular localization.
Reviewer 3 Report
Comments and Suggestions for Authors
See the attached file

English proofreading and grammar revision.
Author Response
Comment 1:
I would like to praise the authors for the substantial revisions made to this manuscript. The title has been clearer and informative. The abstract now provides a good summary of the current work. The methodology section has been strengthened with detailed descriptions of animal housing, dosing, etc. Importantly, the addition of graphical representation of the experimental design, histopathological images (H&E), and immunohistochemistry micrographs greatly improves clarity. The discussion and conclusion have been greatly improved to align with the attained data.
However, I still have minor comments on the figures that could improve your manuscript.
The inclusion of histological and immunohistochemical data is a valuable improvement; however, their current presentation requires refinement to meet publication standards. For Figures 3–5, I would recommend including representative photomicrographs from all experimental groups under study, including controls, to ensure transparency and allow visual correlation with the quantitative results. Furthermore, the addition of negative controls for immunohistochemistry to demonstrate antibody specificity would be helpful.
Response1: We greatly appreciate your valuable feedback and praise for the improvements made to the manuscript. We have revised the figures of the manuscript to include representative images from all histological findings and negative controls.
Comment 2:
Annotate key features: the use of arrows, asterisks, or other markers to highlight key pathological features (ballooning, hemorrhage, leucocytic infiltration, etc) and staining patterns would be more helpful.
Response 2: Thank you for this excellent point. We have now annotated the images with arrows to clearly highlight key pathological features (inflammation, steatosis, fibrosis, ballooning).
Comment 3:
Try to use scale bars on all micrographs, and expand the figure legends, placed directly below each figure (not above), to specify experimental groups, staining method, magnification, and salient morphological or molecular findings (nuclear or cytoplasmic localization). These modifications will significantly enhance clarity, reproducibility, and the overall impact of the visual data.
Response 3: Thank you for your input. The figure legends have now been placed below each figure and expanded to better explain the pathological findings, staining, and magnification. We also included magnification scales on each image.